

# Complet+: a computationally scalable method to improve completeness of large-scale protein sequence clustering

Rachel Nguyen[1], Bahrad A. Sokhansanj[1], Robi Polikar[2] and Gail L. Rosen[1]

[1] Drexel University, Philadelphia, United States of America
[2] Rowan University, Glassboro, NJ, United States of America

## ABSTRACT

A major challenge for clustering algorithms is to balance the trade-off between homogeneity, *i.e.*, the degree to which an individual cluster includes only related sequences, and completeness, the degree to which related sequences are broken up into multiple clusters. Most algorithms are conservative in grouping sequences with other sequences. Remote homologs may fail to be clustered together and instead form unnecessarily distinct clusters. The resulting clusters have high homogeneity but completeness that is too low. We propose Complet+, a computationally scalable post-processing method to increase the completeness of clusters without an undue cost in homogeneity. Complet+ proves to effectively merge closely-related clusters of protein that have verified structural relationships in the SCOPe classification scheme, improving the completeness of clustering results at little cost to homogeneity. Applying Complet+ to clusters obtained using MMseqs2's clusterupdate achieves an increased V-measure of 0.09 and 0.05 at the SCOPe superfamily and family levels, respectively. Complet+ also creates more biologically representative clusters, as shown by a substantial increase in Adjusted Mutual Information (AMI) and Adjusted Rand Index (ARI) metrics when comparing predicted clusters to biological classifications. Complet+ similarly improves clustering metrics when applied to other methods, such as CD-HIT and linclust. Finally, we show that Complet+ runtime scales linearly with respect to the number of clusters being post-processed on a COG dataset of over 3 million sequences. Code and supplementary information is available on Github: https://github.com/EESI/Complet-Plus.

## INTRODUCTION

A critical step in many bioinformatics pipelines is clustering similar nucleotide (DNA or RNA) and protein sequences, such as clustering marker genes that may represent similar taxa or clustering peptides that may have structural homology or similar functions. In biological applications, the number of sequences that must be clustered is extremely large. Annotated protein sequence databases contain hundreds of thousands (of experimentally validated) and tens to hundreds of millions (of predicted) sequences. Metagenome sequencing experiments routinely generate hundreds of millions of reads. As a result,

Corresponding author
Gail L. Rosen, empr3ss@gmail.com

bioinformatics demands scalable clustering methods. Clustering methods generally have a trade-off of increased sensitivity with slower speed, making the protein clustering a challenging problem. Protein clusters are groups of similar (homologous) proteins that most likely share the same or similar function. Clustering procedures must allow compression of information in comparison to the non-clustered representation and may have different resolution levels (*e.g.*, phyla/genus for taxa or superfamily/family for protein).

One of the key challenges in many biological applications is to generate clusters which correspond to the "true" groupings of sequences in the application. For example, if the task is to identify the phyla present in a data set of DNA sequences, ideally clusters should correspond to groups of phyla. If the task is to identify protein families or superfamilies—the focus of this article—then ideally clusters should correspond to protein families. This would then permit clusters to be used for the labeling of unknown sequences. For example, if an unknown protein clusters with proteins from a given family, then the clustering result may be used to predict that the unknown protein is a member of that family. As described above, however, protein sequence data sets are very large, and as a result the sensitivity-speed trade-off requires lowering sensitivity, resulting in relatively tight clusters: clusters that are made up of biologically similar proteins, but with groups of biologically similar proteins, such as a family or superfamily, split up into multiple clusters. In other words, the resulting clusters have relatively high *homogeneity* but relatively low *completeness*.

Optimizing homogeneity over completeness poses a barrier for certain important applications—two examples of which are described here. First, a common use case for clustering is to label otherwise unidentified sequences, such as hypothetical ORFs identified from a metagenomic sequencing data set. Hypothetical ORFs that are most proximate to a cluster, or cluster together with known proteins, can be labeled according to the majority—otherwise, their cluster membership can be determined with consensus. However, clustering methods with low completeness may generate lots of clusters with only one member (singletons) or with only two or a few members—which may not have reliable consensus annotation. Second, clusters can be used to identify and organize groups of sequence homologs, for example to predict functional categories of proteins or groups of proteins with common evolutionary histories. Low-completeness clusters fail in this task as well, particularly as homologs become more remote or when the levels of organization become higher, for example in identifying the members of the same superfamily as opposed to family of protein sequences (*Paccanaro, Casbon & Saqi, 2006*; *Bernardes et al., 2015*). Increasing completeness would require either a slower (more sensitive) clustering method—not feasible for most biological sequence applications—or developing a method to robustly and reliably combine clusters in a way that increases completeness without excess cost in homogeneity. Notably, any decrease in homogeneity means a risk of false positives, where a cluster includes members that are biologically dissimilar to each other.

In this article, we introduce Complet+, a novel method to increase the completeness of clusters obtained using large-scale biological sequence clustering methods. Complet+ addresses a key problem with large-scale clustering methods, such as mmSeqs2 clustering and CD-HIT. Because large-scale clustering tools generally use some kind of iterative or heuristic method, like iterative greedy clustering, they will generally not generate a measure

of identity between representative sequences of clusters. Complet+ utilizes the fast search capabilities of MMSeqs2 to identify reciprocal hits between the representative sequences, which may be used to reform clusters and (1) reduce the number of singletons and small clusters and (2) create larger clusters that better represent superfamilies composed of remote homologs, proteins that share a very distant evolutionary origin. This article begins with a background of the relevant methods and then describes the Complet+ algorithm and the computational experiments used to demonstrate and validate Complet+ on extended Structural Classification of Proteins (SCOPe) families and superfamilies of proteins with structural homology (*Chandonia et al., 2021*). We show that Complet+ can effectively increase completeness with a small or negligible cost to homogeneity on clusters generated using MMSeqs2's default cascade clustering method, singe-step clustering, incremental clustering with MMSeqs2's `clusterupdate` module (*Steinegger & Söding, 2017*), `linclust` (a linear-time clustering method) (*Steinegger & Söding, 2018*), and the popular clustering tool for very large-size sequence datasets, CD-HIT (*Li & Godzik, 2006*). We also show that Complet+ can result in clusters that are far more representative of the true SCOPe family and, in particular, superfamily organization. Finally, we demonstrate that Complet+ is scalable, with linear-time performance with respect to the number of clusters using the same clustering method, and with a manageable cost when used on methods that generate highly similar clusters.

## BACKGROUND

### Sequence clustering

The large scale and complexity of nucleotide and protein sequence data have motivated the development of clustering methods specifically to address these applications. CD-HIT was one of the first such large-scale clustering methods for DNA and protein sequences (*Li & Godzik, 2006*). CD-HIT's essential innovation was to estimate similarity between sequences using k-mers (or short words) rather than performing costly $\mathcal{O}(mn)$ sequence alignment, *via* dynamic programming, or a heuristically-sped up version like BLAST. CD-HIT can be used for a wide range of applications due to its versatility, but one particular application—16S rRNA clustering for *de novo* taxonomic grouping (a.k.a open-reference OTU picking)—immediately became an important application due to the volume of such data being generated (*Chen et al., 2013*). Competing clustering methods soon emerged, each taking a different approach to speed-accuracy tradeoffs, such as UCLUST (*Edgar, 2010*). Historically, however, these tradeoffs have been evaluated to address 16S rRNA sequence clustering, which has particular issues with accuracy that do not apply to protein clustering (*Nguyen et al., 2016*; *Schloss & McMahon, 2021*).

Protein sequence clustering methods are similarly being investigated with an even wider gap between speed and accuracy (*Hauser, Steinegger & Söding, 2016*). There are two categories of clustering algorithms: alignment-based and alignment-free. The traditional alignment-based methods, CD-HIT (*Li & Godzik, 2006*), UCLUST (*Edgar, 2010*), and BlastClust (*NCBI, 0000*) can also be used for proteins in addition to 16S rRNA. Futher developments of Markov Clustering (MCL) techniques were innovated (*Enright, Van*

*Dongen & Ouzounis, 2002*; *Wong & Ragan, 2008*). However MCL techniques, as well as other alignment-based techniques, are computationally expensive. Alignment-free based techniques are emerging, with MMSeqs2 (*Steinegger & Söding, 2017*) being the most popular, and with emerging techniques in deep learning and embeddings (*Karim et al., 2020*). MMSeqs2 clustering is widely used in a range of applications, including clustering known and unknown genes in metagenomes (*Vanni et al., 2022*).

### MMseqs2

Many-versus-many sequence searching (MMseqs2) is a software suite that offers a wide array of tools for sequence alignment (*Steinegger & Söding, 2017*) and clustering (*Steinegger & Söding, 2018*; *Hauser, Steinegger & Söding, 2016*) of both nucleotide and protein sequences. It has high sensitivity and is efficient in both computation time and hardware resources, as it is optimized for multi-threaded use. It is also highly scalable and excels at working with large datasets. In this study, we examine MMseqs2's sequence alignment and clustering tools.

Sequence alignment (*Steinegger & Söding, 2017*) is performed using MMseqs2's `search` module, and is comprised of three steps: a k-mer match step, a vectorized ungapped alignment, and a gapped (also known as Smith-Waterman) alignment. The k-mer match step drastically reduces the number of Smith-Waterman alignments, and is a main contributor to the `search` module's high speed.

A search is characterized by many criteria and parameters, one of the more notable ones being its sensitivity. A search run with a low sensitivity setting will be fast, but it will find less hits than a search using a high sensitivity setting. The `search` module also offers the ability to restrict the output to alignments that are within a specified significance threshold (also known as expectation-value, or e-value).

Two major modules within MMseqs2 for clustering are: `linclust`, a linear method, and `cluster`, a cascaded method that begins with `linclust` before performing additional prefiltering, alignment, and clustering. Both clustering modules are unsupervised and their results are independent of the order sequences are presented in. (MMseqs2 further offers the option of simple clustering in a single step without cascading, but that is not its default operation.)

MMseqs2's implementation of linear clustering is the `linclust` module (*Steinegger & Söding, 2018*). The computational runtime of `linclust` scales linearly with the number of input sequences. The `linclust` algorithm begins by grouping sequences that share a k-mer, selecting the longest of sequence of each group as the center (or representative sequence). Every sequence is then compared(using Smith-Waterman alignment) to every representative it shares a k-mer with, and if it passes the clustering criteria it is recruited into the cluster.

The cascaded clustering module is the default setting of the `cluster` module in MMseqs2 (*Steinegger & Söding, 2018*). The cascaded clustering method first runs the `linclust` module to produce an initial clustering. It then performs a prefiltering step followed by a Smith-Waterman alignment of the sequence pairs that passed. Next, an initial clustering with low sensitivity and high significance threshold is done, which results

in a quick clustering that only yields matches with high sequence identity. It then repeats the process beginning at the prefiltering step, instead using the first step's cluster representative sequences rather than all sequences. It also uses a higher sensitivity than the first step. The process is repeated a third time using the sensitivity specified by the user, and the results are finally merged. Cascaded clustering can be performed with as many as seven steps, but three is the default.

The `cluster` module's sensitivity setting is similar to search's sensitivity setting: a low sensitivity clustering will be faster than one at the highest sensitivity, but the result will be comprised of many more clusters due to finding less hits. The `cluster-reassign` option will recompute the clusters' representative sequences at each clustering step. Otherwise, `cluster` will use the representative sequence initially chosen for each cluster at each step.

The `cluster` module's `single-step-clustering` option changes the workflow to perform the prefiltering, alignment, and clustering of the sequences using the user's specified criteria at the first and single step, as opposed to gradually over the course of three steps like `cluster`.

## Incremental learning and the cluster updating workflow in mmSeqs2

Incremental learning is a developing approach to handling the rapid growth of massive amounts of biological sequence data, particularly in the era of next-generation sequencing. Classically, a database needs to be *de novo* clustered or a supervised machine learning classifier needs to be fully retrained each time a sequence database is updated with new examples. An incremental approach can instead learn fundamental features and classes of the data continuously as new examples are added (*Zhao, Cristian & Rosen, 2020*). Both supervised and semi-supervised methods have been proposed for classifying sequences (*Ozdogan et al., 2021*; *Zhao, Cristian & Rosen, 2020*; *Dash et al., 2021*). One incrementalization approach is to reduce the time required for sequence alignment by saving information and running only on new increments (*Dash et al., 2021*). Another is to incrementally update gene/genome classifiers, such as NBC++ (*Zhao, Cristian & Rosen, 2020*) and Struo2 (*Youngblut & Ley, 2021*). Our group has begun to explore incrementalization methods for unsupervised methods that update the cluster representatives, and semi-supervised methods that rely on clustering information in combination with learning classifiers for both taxa and protein sequence classification (*Halac et al., 2021*; *Ozdogan et al., 2021*).

MMseqs2 (*Steinegger & Söding, 2018*) is one of the most commonly used software suites for protein sequences in academic and commercial applications. It is the only one that offers incremental cluster updating. The `clusterupdate` module allows the user to update a previous clustering result following the addition or removal of sequences. When updating the clustering result, the module keeps the sequences' identifiers consistent within the original clustering's databases. It also shares many of the settings the cluster module offers, most notably the sensitivity setting. The `clusterupdate` module first compares the new sequences to existing ones with a supervised process. Then, when sequences which are not similar enough to any of the existing clusters are added, they are compared to each other in an unsupervised process, with some becoming representatives of new clusters.

## METHODS

All tests were performed utilizing a single CPU thread and 16 GB of memory.

### Protein data sets

#### Structural classification of proteins (SCOP)

Developed by MRC Laboratory of Molecular Biology, the SCOP database classifies proteins first by their secondary structure, followed by their evolutionary relationships primarily through manual curation until work on it ceased June 2009 (*Hubbard et al., 1997*). Its development was continued with the release of SCOP—extended (SCOPe) database (*Fox, Brenner & Chandonia, 2013*) in March 2012 by Berkeley Lab and UC Berkeley, who continue to expand it with a combination of manual curation and automated methods. The data itself are the genetic domain sequences of proteins within PDB SEQRES records.

As of version 2.08, SCOPe contains 302,566 protein domain sequences in total. The topmost level of classification is sequence class, followed by fold, superfamily, and lastly family. SCOPe 2.08 is encompassed by seven classes, 1,257 folds, 2,067 superfamilies, and 5,084 families (*Chandonia et al., 2021*). For this study, we only evaluate the clustering results on the superfamily and family levels of classification.

#### Clusters of orthologus genes (COG)

The COG database (*Galperin et al., 2019*) groups protein functional domains into COG's according to their orthologous relationships. Each COG thus represents a general function. For example, glutamyl- or glutaminyl-tRNA synthetase proteins are classified under COG0008. Each of these COGs are further defined by one or more functional groups, there being 26 functional groups total. The current release as of May 12, 2022 includes 3,456,089 functional domains encompassed by 4877 COGs (*Galperin et al., 2021*).

#### Swiss-Prot COG

The Swiss-Prot database (*Bairoch & Apweiler, 1999*) is a subset of the Uniprot. The database is composed of manually-annotated records with information extracted from literature and curator-evaluated computational analysis. We filter the COG annotations using only these manually-annotated proteins to obtain biologically-verified COGs. The Swiss-Prot COG subset contains 122,202 functional domain sequences encompassed by 3,858 COGs.

### The Complet+ algorithm

The Complet+ algorithm builds on unsupervised clustering results by aligning cluster representatives (*via* MMSeqs2 search) and merging clusters that are reciprocal hits within a particular e-value. In the results, we show that Complet+ can be used on any base algorithm, thus it implements "incremental learning" without needing all previous data (just cluster representatives and a mapping of sequences to clusters), and it runs in less time than rerunning a more sensitive clustering algorithm from scratch. Also, currently, there are no other postprocessing algorithms that improve completeness without having to be rerun, from scratch, on all the data. Therefore, Complet+ fills a niche in optimizing clustering for completeness in an incremental learning manner.

Complet+ takes any (1) FASTA file of representative (or all) sequences and (2) clustering algorithm results as input after they have been converted to a particular TSV format, which is a simple two-column data file of sequence IDs and the IDs of their cluster representatives. For example, MMseqs2 results are obtained from MMseqs2's `createtsv` command, but Complet+ is compatible with any clustering data file regardless of its source, provided it is converted into the same format. The FASTA file of the sequences only needs to contain the sequences of the cluster representatives, but if there are additional sequences, the algorithm extracts the relevant ones. Representative sequence extraction is performed through the use of grep, a command-line tool used for searching patterns within files. The representative sequences, as described in the cluster data file's first column, are all extracted to a new FASTA file. The clustering data file is used again in the relabeling step (Fig. 1).

With this FASTA file of representative sequences, Complet+ then performs a pairwise sequence alignment between all representatives by using the MMseqs2 search function, with the pairwise combinations as query and target arguments. We found that the homogeneity-completeness was optimized at the highest sensitivity option, $s = 7.5$, of the MMseqs2 `search` function. As shown in the Cascade-C+ results in the Results and Discussion Section, a Complet+ `search` sensitivity of 7.5 ($s = 7.5$) increased the Default completeness by 5.99% (while lowering homogeneity by 0.05%), while a Complet+ `search` sensitivity of 5.7 ($s = 5.7$) only increased the Default completeness by 3.63% (while lowering homogeneity by 0.02%). The other settings, such as e-value threshold and query-coverage, were left at the default values of 0.001 and 0.8, respectively. Also, Complet+ can even improve upon Cascade Clustering and Connected Component (CC) clustering with the max sensitivity of 7.5. The algorithm employed by Complet+ is similar to connected component clustering (cluster mode 1 in MMSeqs2), but has an even looser constraint to improve completeness. The looser constraint is that instead of connecting nodes that are best reciprocal hits by raw score, Complet+ looks for all reciprocal hits that fall within a user-defined e-value. From the reciprocal hits, we are able to deduce which clusters are similar to each other, in order to merge them *via* the following described process.

After the pairwise alignments of the cluster representatives are obtained, the alignment results are sorted from lowest e-value to highest and then filtered, retaining only sequences that are reciprocal hits within the specified e-value to the algorithm. Reciprocal hits are pairs of sequences where one is in the top hits of the other and vice-versa when queried against all sequences (in this case, all the cluster representatives). Alignment results that are self-hits (where the query and target sequences are the same sequence) are filtered out. Figure 2 depicts how only the reciprocal hits are retained following the alignment filtering step. It is important to note that the reciprocal hits need not be the "reciprocal best hits", or best-scoring hits for the sequences. A sequence can and likely will have multiple sequences to which it is reciprocal.

We observed that the reciprocal hits obtained through this method formed clusters of their own. A given representative sequence would usually have between two and six reciprocal hits, following the sequence alignment. Those reciprocal hits would tend to only be reciprocal hits to each other. Figure 3 visualizes these reciprocal-hit results and the clustering observed. As evident in the visualization, these reciprocal hits form numerous
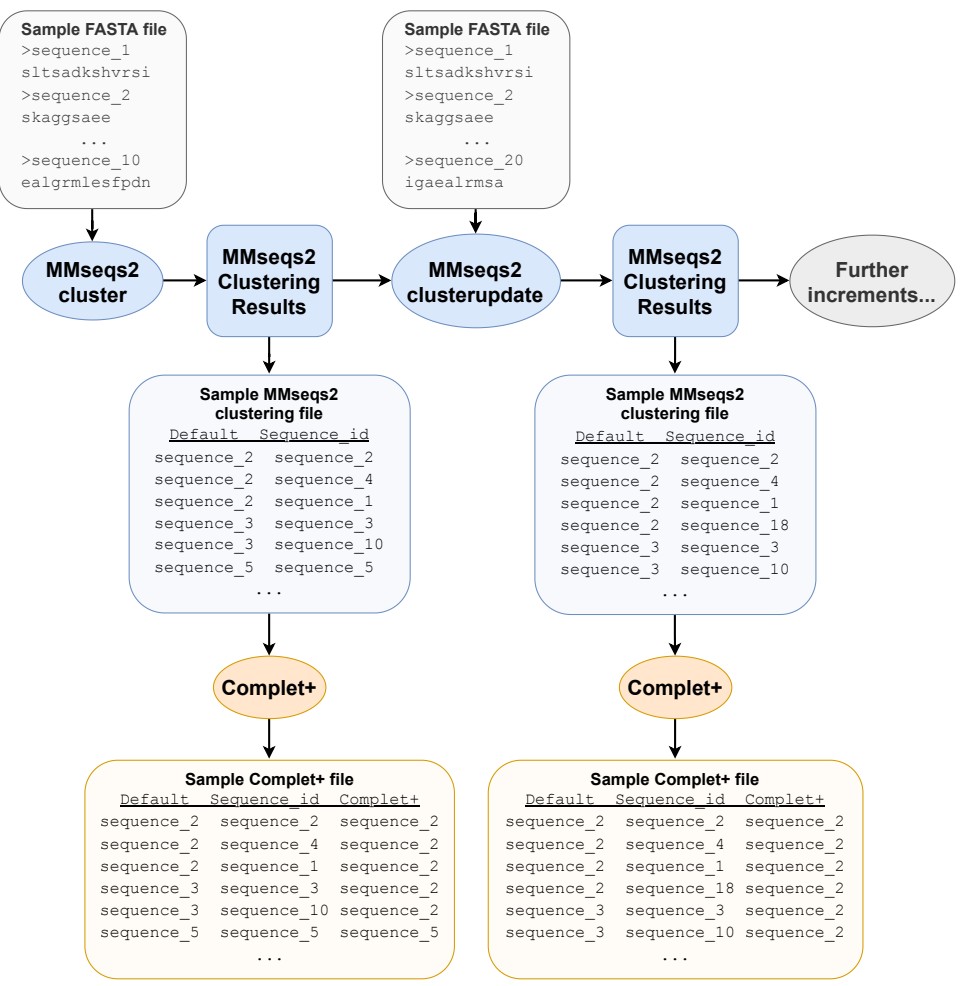

**Figure 1   Example pipeline depicting Complet+ usage.** Not all files required and produced by the MM-seqs2 tools and Complet+ are depicted, merely the most prominent ones, for clarity. The pipeline depicted only includes one clusterupdate module use following the initial clustering with the cluster module, but the tests discussed later feature more successive increments with clusterupdate.

isolated groups akin to clusters. We relabel the members of these clusters with the ID of one of the members in order to merge them.

Beginning with the first representative sequence in the file, all subsequent representative sequences with which a sequence is reciprocal are relabeled as the original representative sequence. Complet+ then iterates through each of those reciprocal hits' own reciprocal hits, which are also relabeled as that first sequence. Complet+ traverses through the data in this recursive fashion, ignoring representative sequences that have already been passed. All of the members of that original representative sequence are hence identified and relabeled. Complet+ repeats this process for every representative sequence in the file, ignoring those that have already been seen. This process results in a list of new representative sequences which each former representative sequence and its cluster members are relabeled to. The

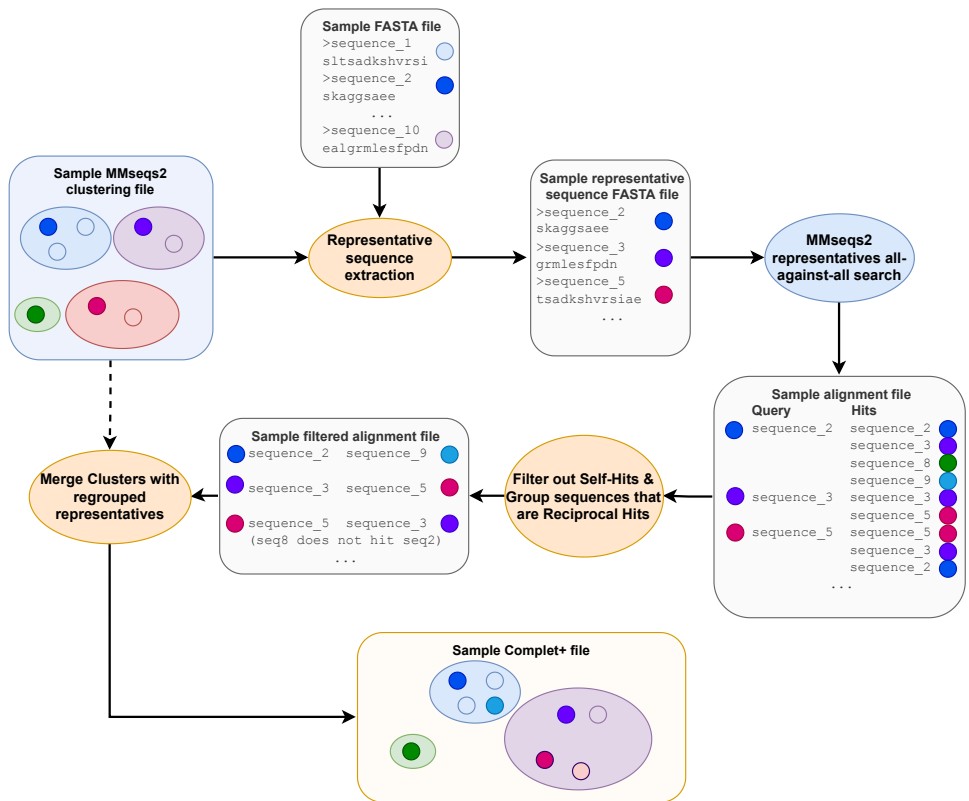

**Figure 2** **The Complet+ algorithm.** The alignment files contain additional data columns, including the one containing each alignment's e-value (not depicted) that is used in the alignment sorting and filtering step.

old cluster ID and Complet+ cluster ID, corresponding to each sequence ID are then saved into the Complet+ output, seen in Fig. 1.

The algorithm then passes through the data file of the MMseqs2 clustering results. If a sequence's representative is on the aforementioned relabeling list, that sequence is relabeled with the new representative sequence ID. The resulting clustering data file contains the data from the original clustering file, in addition to the new cluster representative of each sequence (Table 1).

## Clustering metrics

To evaluate the performance of both MMseqs2 and our post-processing algorithm, we used Scikit-learn's cluster metrics library (*Pedregosa et al., 2011*) to calculate homogeneity, completeness, adjusted mutual information score (AMI), and adjusted Rand index (ARI).

### Homogeneity and completeness

Homogeneity and completeness are two metrics often used together—and can also be combined into a V-measure—to evaluate clustering results (*Rosenberg & Hirschberg, 2007*). Homogeneity is a measure of a clustering purity: a clustering result has perfect homogeneity (of 1) if all members of any given cluster are truly of the same class. Homogeneity score is

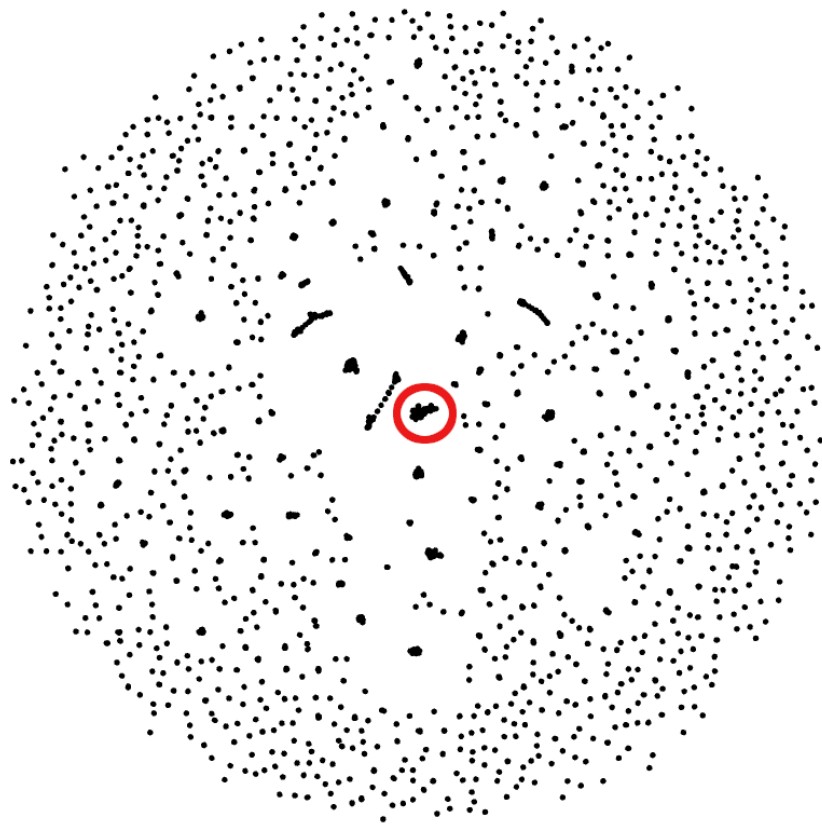

**Figure 3  Gephi visualization of the MMseqs2 search of the representative sequences.** Depicted are the reciprocal hits resulting from the sequence alignment. Each dot represents a representative sequence. Dots in close proximity indicate a reciprocal hit. Groups of close reciprocal hits are effectively clusters. Complet+ merges these clusters to improve completeness. Circled in red is one such cluster. Every representative sequence within this circle is a reciprocal hit to at least one other representative sequence in the circle. Complet+ merges these representative sequences, resulting in a single cluster Every single sequence in the image has a reciprocal hit; a dot that appears to be a single sequence is actually two or more dots on top of each other.

penalized (reduced) if its clusters contain members that are of different classes. A clustering result has perfect completeness (of 1) if all members of each class are within the same cluster. Completeness score is penalized (reduced) if any class members are split amongst multiple clusters. Finally, V-measure is the harmonic mean between homogeneity and completeness and is also known as the normalized mutual information metric (*Rosenberg & Hirschberg, 2007*; *Pedregosa et al., 2011*).

### Adjusted Mutual Information (AMI) and Adjusted Rand Index (ARI)

The mutual information score (MI) is an information theoretic measure of agreement between two clusterings. In our case, the two clusterings are the true clusters (as determined by ground-truth classes) and the predicted clusters. The AMI is obtained by correcting MI for chance (*Vinh, Epps & Bailey, 2009*). The Rand index (RI) is a measure of similarity between two clusterings. ARI is the RI adjusted for chance (*Vinh, Epps & Bailey, 2009*).

**Table 1** **The output of Complet+.** The file retains the representative sequence ID for each sequence's old cluster label, allowing simple comparison between the old and new labels.

| Old cluster ID | Sequence ID | Complet+ cluster ID |
|---|---|---|
| d1u9ca_ | d1u9ca_ | d1u9ca_ |
| d1u9ca_ | d1pv2c_ | d1u9ca_ |
| d1u9ca_ | d1pv2d_ | d1u9ca_ |
| d1n57a_ | d1n57a_ | d1u9ca_ |
| d1n57a_ | d1izya_ | d1u9ca_ |
| d1n57a_ | d1izza_ | d1u9ca_ |

While superficially similar, AMI and ARI in fact reflect different relationships between predicted clusters and ground-truth classifications depending on whether the clustering solutions are balanced or imbalanced. A balanced clustering solution is one of equally-sized clusters, whereas an imbalanced solution's cluster sizes vary greatly. An imbalanced solution is more likely to present clusters that are pure, due to likely having more small clusters than a balanced solution. Specifically, AMI tends to give higher scores to (or is biased towards) unbalanced clustering solutions, while ARI is biased towards those that are balanced (*Romano et al., 2015*).

## RESULTS AND DISCUSSION

A primary goal of Complet+ is to serve as a versatile approach that can be used with any clustering algorithm. Therefore, we evaluated Complet+ on a variety of initial clustering algorithms. We also evaluated Complet+ on both classical single-batch analysis and the incremental multi-batch analysis used for incremental learning applications: specifically, single-batch clustering tests, two 5-batch incremental learning tests, and a 50-batch incremental learning test.

In single-batch clustering, Complet+ is run after the SCOPe dataset is clustered with a single cluster call (using variety of methods provided by mmSeqs2, as well as CD-HIT). Complet+ was then run on the resulting clustering output. In addition, two batch partitioning test data sets were created to help further evaluate the performance and scalability of the method as described below: (1) for each of the five batches, ~20% of the new classes (previously unseen by the clustering method) were added to each batch, and (2) each of the sequences in the five batches were randomly chosen. Both tests result in five folds of approximately equal number of proteins. To evaluate incremental learning applications, we create a simulated scenario by inputting the first batch to MMseqs2's default cascaded `cluster` module, then sequentially adding each of the remaining four groups to the database using MMseqs2's `clusterupdate` module. Complet+ is then run on the MMseqs2's clustering results from each of these five steps. Using a cross-validation approach, this is repeated an additional four times, for a total of five folds. For the new classes and random partitioned data, the clustering metrics were averaged (and standard deviation was calculated) across the five folds. Clustering metrics were calculated between predicted labels (the clustering result) and true labels (SCOPe family/superfamily).

## Complet+ improves clustering metrics
### Applying Complet+ to MMseqs2 clustering and CD-HIT

We first show that Complet+ significantly reduces the number of singletons and small clusters that are formed. Figure 4 shows the reduction in the number of small clusters when applying Complet+ to the results of mmSeqs' single-step clustering on the SCOPe data set. There is a significant reduction in the number of singletons, and consistent reductions of up to 50% in the numbers of clusters with few representatives. While not shown on the graph, Complet+ also generates larger clusters than default methods.

Overall, CD-HIT produces numerous clusters, clustering the 303,000 SCOPe sequences down to about 204,000 clusters. The proliferation of clusters is evident in CD-HIT's low completeness score shown in Fig. 5. MMseqs2 generated fewer clusters than CD-HIT, its `linclust` module producing about 30,000 clusters, and the cascaded clustering tests generated between 9,000 and 10,000 clusters. These results are shown in Figs. 5A and 5B. All default results had excellent homogeneity scores as well, at or above 0.95.

Figures 5A and 5B show the homogeneity/completeness, AMI, and ARI for the single-batch tests using MMSeqs2 and CD-HIT. For all tests, Complet+ was able to substantially improve completeness without measurable loss in homogeneity at the superfamily level and minimal loss at the family levels of classification. The composite of homogeneity and completeness, also known as the V-measure (not shown in the graph but but can be calculated from the homogeneity and completeness) is substantially higher when applying Complet+ at the superfamily-level when averaged across all single-batch tests: 0.80 *vs.* 0.89. The increase in V-measure is less pronounced but still favorable at the family-level: 0.86 *vs.* 0.91. As indicated by Figs. 5A and 5B, the V-measure increase is due to the homogeneity of the default Complet+ clustering remaining mostly unchanged, while completeness increasing.

The lower homogeneity of both Default and Complet+ results at the family *versus* superfamily level are expected. There are many more families than superfamilies. If there is a mis-clustering at the superfamily-level, it will be reflected on the family level as well, but not vice versa. On the other hand, completeness is higher on the family level because there are more families than superfamilies. Both Default and Complet+ are conservative with their clustering, having a tendency to create multiple smaller clusters over large ones, which then results in lower completeness. Furthermore, since the superfamily is the larger clade above family, the score is more penalized when evaluated on the superfamily level.

As shown in Figs. 5A and Fig. 5B, Complet+ also results in an increase in both AMI and ARI when compared to the Default clustering. Regardless of how low the Default AMI score was, Complet+ was able to improve it to a score above 0.8. The improvement to ARI was also notable across most test cases, where Complet+ scores were higher relative to the Default scores. For both Default and Complet+, AMI was notably higher than ARI in most cases, which was expected due to AMI's previously discussed bias towards datasets with unbalanced ground-truth classes.

We can see from the plots in Fig. 5 that while there is improvement in completeness over connected clustering, the overal AMI/ARI are quite similar. In fact, for family-level, the difference in AMI is only by 1%. Upon further investigation, this small yet overall
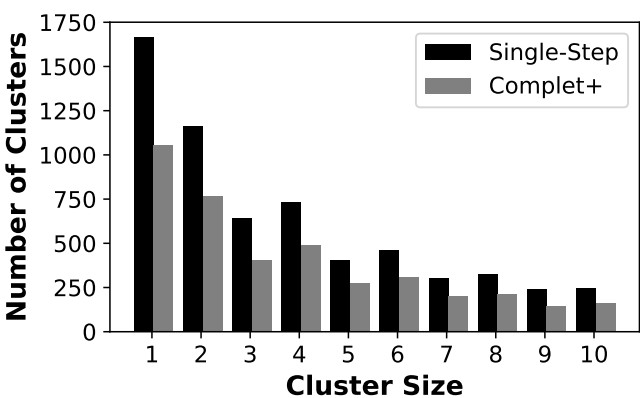

**Figure 4** Number of clusters having one (singletons) through ten members for the MMSeqs single-step clustering before and after applying Complet+.

In the first row, ●: Default Homogeneity, ●: Default Completeness, ●: Complet+ Homogeneity, ●: Complet+ Completeness.
2nd and 3rd rows: ●: Default, ●: Complet+

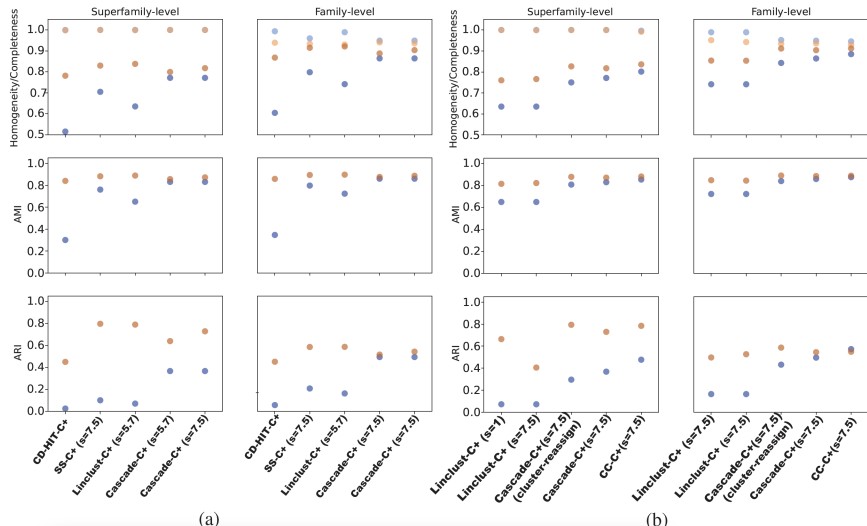

**Figure 5** The homogeneity, completeness, AMI, and ARI of the single-batch tests. (A) All tests aside from "CD-HIT" use MMseqs2 the clustering module stated. The sensitivity specified in parentheses refers to the MMseqs2 search run by Complet+, not the clustering sensitivity, which was the default value of 4.0 where applicable. Overall, Complet+ substantially improves each test case's completeness at little expense to homogeneity. The loss in homogeneity is more notable when evaluating the clustering results on the family-level of classification, however still to a lesser degree than the increase in completeness. Complet+ also improves the AMI and ARI of each clustering to varying degrees, having a generally greater improvement. (B) The two leftmost tests are Complet+ run at minimum, and maximum MMseqs2 search sensitivity, each on the same Default linclust test results. The two following tests are identical aside from one using the cluster-reassign setting. The last test is the Connected Component (CC) clustering method of MMSeqs2, run at the highest sensitivity. Like the results in (A), AMI and ARI are improved with each case while completeness is also improved without significant loss of homogeneity.

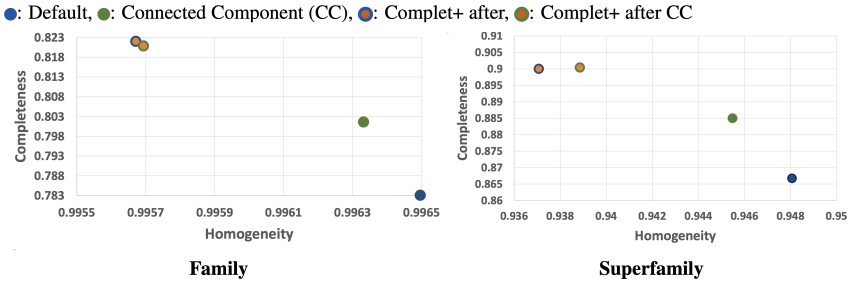

**Figure 6** **Homogeneity/Completeness scatterplots demonstrating base-algorithm + Complet+ *vs.* base algorithms (Default, Connected Component clustering both with highest sensitivity of 7.5).** Complet+ can improve completeness of each algorithm by a greater amount than is lost in homogeneity. By merging clusters whose representatives are reciprocal hits (given an e-value threshold) of each other allows more clusters to be merged than stricter connected node criteria by the CC algorithm.

improvement can be seen for all sensitivity levels, as shown in the connected component with Complet+ supplementary graph on the Github page (https://github.com/EESI/Complet-Plus/blob/main/figures/connected_component_study/cc_fam_superfam.pdf). Examining the homogeneity/completeness in scatterplot form, Fig. 6 illustrates that improvement in completeness is still a few percentage points at the cost of a fraction of a percentage point in homogeneity for even the highest sensitivity of 7.5. Interestingly, for sensitivity of 4 (in the supplementary graphs, https://github.com/EESI/Complet-Plus/blob/main/figures/connected_component_study), the default+Complet+ is similar in performance to connected component clustering—however, improved completeness can be gained by running Complet+ on connected clustering. While these algorithms are similar, due to the incremental learning nature of Complet+, it can usually make some improvements given general similarities of the cluster representatives.

Finally, we can see that the improvements in completeness are notable but overall improvement is less at higher sensitivities. The runtime of both the default MMseqs2 and Complet+ significantly increase as the sensitivity increases, as shown with the connected component (CC) clustering runtimes *vs.* sensitivity for the two algorithms shown in Fig. 7.

### Batch incremental learning using MMseqs2 `clusterupdate`

The results of incremental tests are consistent with the single-batch tests. As Fig. 8 shows, applying Complet+ results in an increase in completeness significantly greater than any concomitant reduction in homogeneity for incremental clustering (*i.e.,* V-measure is higher) as well. Notably, in Fig. 8, the new classes (class-partitioned) results have a higher variance than that of the random partitioning. This is due to the difference in the data partitioning; in the case of the new classes test, 20% of the classes are added in each batch. In the case of the randomly-partitioned test, 20% of the data is added in each batch so the algorithm has seen the vast majority of the classes within the very first batch. MMseqs2 performs best when clustering incoming sequences if their true class is already represented in the database. This is consistent with the low variance seen in the metrics of the randomly-partitioned results. In the class-partitioned test, it is possible that some
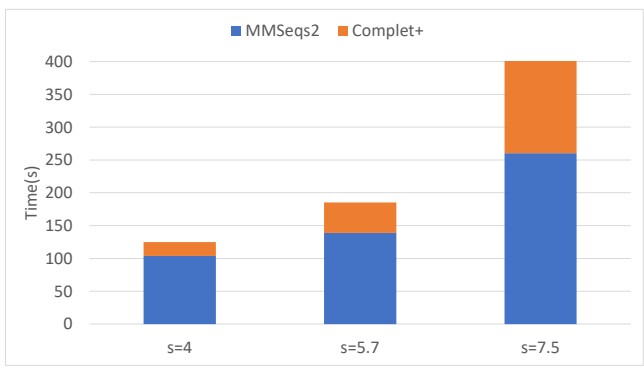

**Figure 7  Algorithm runtime *vs*. sensitivity levels for MMSeqs2 and Complet+.** Both algorithms' run-times increase polynomially.

incoming classes may be similar to classes that have already been seen. Such sequences may then be clustered with existing clusters rather than being placed into new clusters. The batch that these clusters are in will vary between the tests, potentially causing increased variance in clustering metrics.

For the class-partitioned incremental tests, the metrics of Batch 1 clustering were lower than those of the single-batch clustering. With each successive batch, homogeneity/completeness, and AMI increased, and by the last batch they were nearly as high as the metrics of the single-batch test. ARI slightly decreased with successive batches indicating a bias in clustering sequences together rather than making false positives/negatives randomly (which ARI rewards). As previously explained, AMI is the better measure for ground truth clusterings that are unbalanced, and ARI is better for balanced ground truth clusterings. Both SCOPe and Swiss-Prot COG are unbalanced datasets and thus favored by AMI. With the random tests, the metrics of the MMseqs2 `clusterupdate` algorithm never reached the single-batch metrics—however, Complet+ relatively stayed the same. The homogeneity and completeness of both the new class and random tests were relatively close in value by the final batch.

To further evaluate the potential applicability of Complet+, we also tested it on the aforementioned MMSeqs2 clustering methods and CD-HIT on the Swiss-Prot COG database. Full results are provided at our GitHub site for this article, (https://github.com/EESI/Complet-Plus.) In brief, when applying Complet+, the Swiss-Prot COG (only the Swiss-Prot portion of the COG database) class-partitioned test saw similar improvements in V-measure across the board. However, single-batch and random-partitioned tests do not show the same level of improvement. While Complet+ increased the completeness of the Default clusterings, the losses in homogeneity were greater than those observed in the SCOPe tests. This result is likely due to nearly identical sequences being classified in different COGs. Accordingly, schemes like COG, which classify sequences apart despite a high degree of similarity, are not well suited for Complet+.

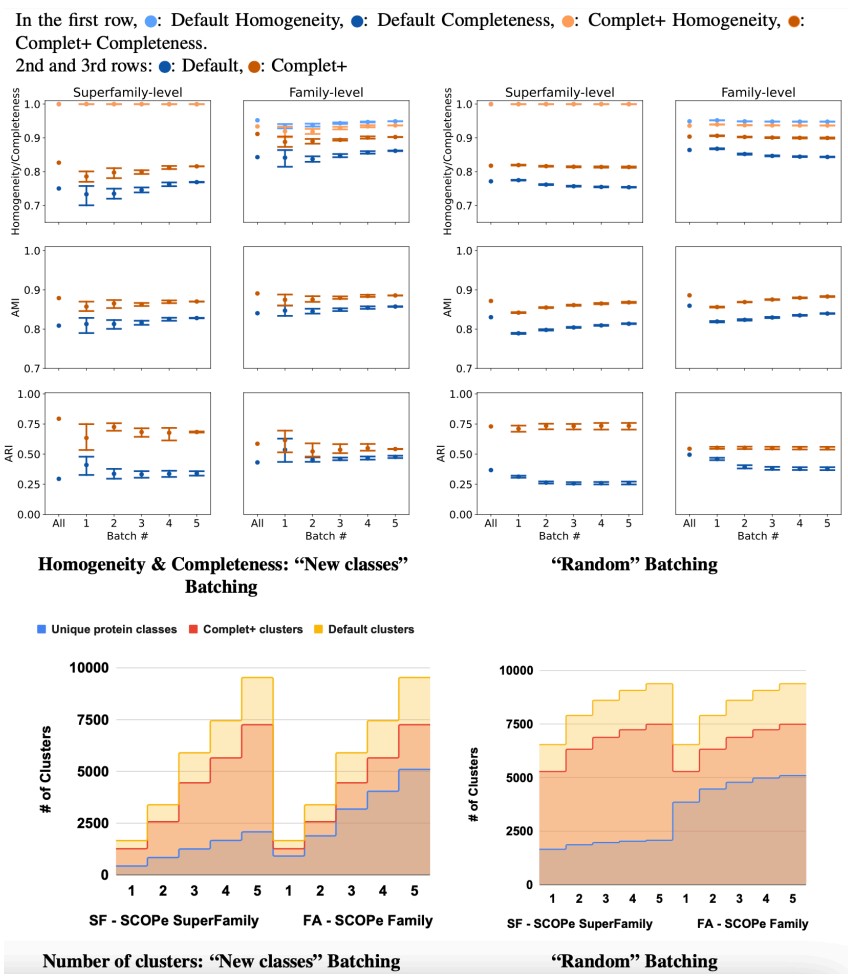

**Figure 8** Homogeneity, completeness, AMI, and ARI of the superfamily- *vs.* family- level for both (A) "new classes" and (B) "random" test batching partitions (for five batches) for MMSeqs `cluster` and `clusterupdate` followed by Complet+. The tick labeled "All" on the graphs represents clustering all sequences in one single batch. Overall, Complet+ increases MMSeqs2 completeness by substantially more than it reduces homogeneity relative to the default MMSeqs2-generated clusters. Using Complet+ results in an increased AMI and ARI at both family and super-family levels. Also, we can see that discovery of new classes yield a large variance in performance as opposed to the base algorithm obtaining most classes in the first batch. The variance is due to the number of actual families or super-families ("true" clusters). (C & D) Number of true and predicted clusters for default MMSeqs2 and Complet+. The number of true clusters is always lower than what Default MMSeqs2 finds, and Complet+ is able to reduce them 10–20% by merging proteins that belong to the same family/superfamily.

## Complet+ is computationally scalable

Batch learning, which is a type of incremental learning, demands that an algorithm should scale linearly with increasing data. We therefore ran several types of experiments to examine the runtime performance of Complet+. Taking the runtimes of all the different modes in SCOPe from (Figs. 5A and 5B), we plotted the run times *vs.* # of clusters input into the algorithm, shown in Fig. 9. We fit a power law to the curves, and SCOPe runtime-inputclusters follows an $x^{1.36}$ while the Swissprot COG curve follows $x^{1.2}$. While it can be

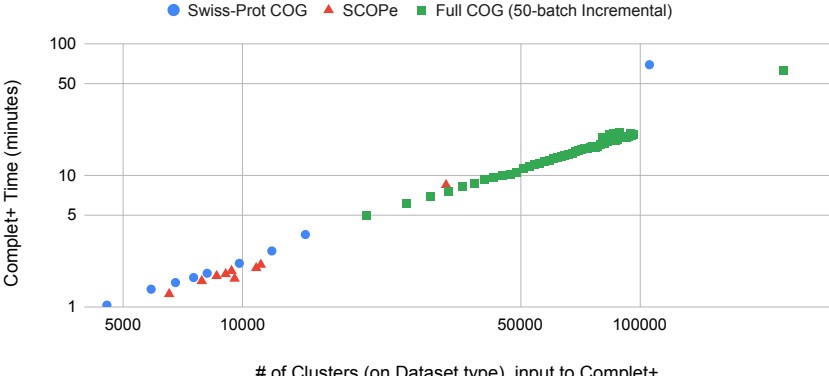

**Figure 9** **The Complet+ time *vs.* the number of clusters produced by a variety of algorithm modes in MMseqs2.** Some algorithms deviate from the line on the log–log plot due to the output cluster representatives and the different relationships between them (*e.g.*: cascaded tends to produce more less similar representatives while `linclust` produces more similar ones). Using the same type of algorithm on the 50-batch large dataset, Complet+ scales linearly *vs.* the number of input clusters.

interpreted that Complet+ runs polynomial in time as a function of the number of input clusters, there is additional computational complexity. The time is due to the number of input clusters as well as the representative sequence similarity. If more representative sequences are similar than other clustering algorithms, the search will take more time. This is the case with the furthest blue and red point along the *x*-axis, which are due to the algorithm `linclust`. `linclust` yields cluster representatives that are more similar to each other, and therefore, adds another variable in the time complexity. We then conducted another experiment that considered the large dataset of 3 million proteins from the COG database shown in Fig. 10. After splitting it into 50 batches and **only** using the `clusterupdate` algorithm, the curve (in green) fits a $x^1$ power law which is exactly linear in time to the number of clusters. Therefore, if the same underlying algorithm is used, Complet+ is linear in time given the number of input clusters; otherwise, the relationship between representative sequences also needs to be taken into account between the algorithms.

Complet+ demonstrates linear time performance on the 50-batch incremental experiment on the full COG database in Fig. 10. Because the current implementation of Complet+ can take 80% to 175% of the original MMseqs2 `cluster` time, we questioned the scalability of the method. As mentioned, the time that Complet+ takes is a function of how many clusters that the original algorithm (*i.e.,* MMseqs2 `clusterupdate`) produces, and the distances between cluster representative sequences. Also, cascaded clustering produces many fewer clusters than `linclust` or single-step clustering, and therefore, running Complet+ halves in time. Complet+ takes less time than the default cascaded algorithm, shown in the single-batch "All" bar of Fig. 10. Complet+ does take longer than `clusterupdate` on all sequences. However, in incremental learning settings, where batches of sequences are being incrementally added, Complet+ is shown again in Fig. 10

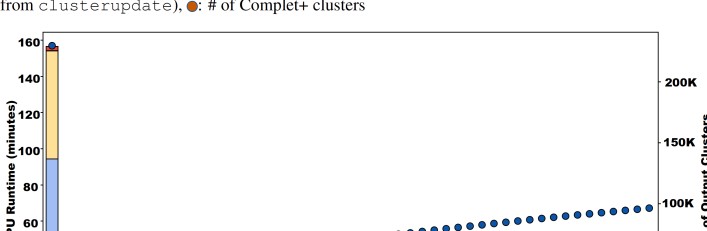

**Figure 10  Runtimes of MMseqs2 cascade clustering of sequences in a single batch and 50 increments of MMseqs2 clusterupdate, shown for a 3 million protein sequence data set from the COG database, with and without running Complet+ at each step.** While Complet+ takes significantly longer than MMseqs2 clusterupdate, it scales linearly to the number of input clusters.

(in addition to the green curve in Fig. 9) to increase linearly with the amount of input clusters. Therefore, Complet+ is scalable to large datasets in an incremental setting.

## CONCLUSIONS

Complet+ offers a transparent and easy-to-use solution for merging clusters to improve completeness and reduce the number of potentially redundant, similar clusters. Complet+ demonstrates consistent improvements to completeness with low penalty to homogeneity, all the while possessing linear scalability. Running Complet+ substantially reduces the number of singletons and very small clusters, which is critical for improving the performance of unsupervised methods for biological analysis. Complet+ also produces more biologically representative clusters: achieving a substantial increase in AMI and ARI metrics, which compare predicted clusters to biological classifications. Complet+ is also a versatile tool, improving clustering metrics on clusters generated using a wide range of algorithms, including MMseqs2's single-step, cascade clustering, connected component clustering, and `clusterupdate` modules, as well as `linclust` and CD-HIT. Finally, we show that Complet+ is linearly scalable with respect to the number of clusters being post-processed, by testing it on a COG dataset of over 3 million sequences. The software and source code are available at https://github.com/EESI/Complet-Plus.

## ACKNOWLEDGEMENTS

We would like to thank Drexel University's University Research Computing Facility (URCF) for providing hardware where some of the computations were run. We thank Evan Yan for helping with the command line interface and compiling the Complet+ Docker container.

### Funding
This work is supported by NSF grants #1936791, #1919691, #1936782, and #2107108. The funders had no role in study design, data collection and analysis, decision to publish, or preparation of the manuscript.

### Grant Disclosures
The following grant information was disclosed by the authors:
NSF grants: #1936791, #1919691, #1936782, #2107108.

### Competing Interests
The authors declare there are no competing interests.

### Author Contributions
- Rachel Nguyen conceived and designed the experiments, performed the experiments, analyzed the data, prepared figures and/or tables, authored or reviewed drafts of the article, and approved the final draft.
- Bahrad A. Sokhansanj conceived and designed the experiments, prepared figures and/or tables, authored or reviewed drafts of the article, and approved the final draft.
- Robi Polikar conceived and designed the experiments, authored or reviewed drafts of the article, and approved the final draft.
- Gail L Rosen conceived and designed the experiments, analyzed the data, prepared figures and/or tables, authored or reviewed drafts of the article, and approved the final draft.

### Data Availability
The code is available at Github: https://github.com/EESI/Complet-Plus;
rtn28, gailrosen, & ev4nyan. (2022). EESI/Complet-Plus: Complet-Plus v1.0 (1.0). Zenodo. https://doi.org/10.5281/zenodo.7449402.

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
