# Peer review of "Complet+: a computationally scalable method to improve completeness of large-scale protein sequence clustering"

_PeerJ, doi:10.7717/peerj.14779_

## Round 0.1 · original submission · Major Revisions

Dear Dr. Rosen

Thank you for submitting your manuscript for review to PeerJ - the Journal of Life & Environmental Sciences

After careful consideration, we feel that your manuscript will likely be suitable for publication if it is revised to address all the points below. Therefore, my decision is "Major Revision."

We invite you to submit a revised version of the manuscript that addresses the points specified by the 3 referees.

·

Basic reporting

The authors did outstanding work. They implemented method with python to sequence clustering which improves homologs groups and completeness. It’s an incremental work to do post-processing of mmSeq2 and CD-HIT. Authors beautifully utilized MMSeq2 reciprocal hit search method.
I have a few concerns which are as follows:
1. The authors have used 16S rRNA clustering problem (line:105); make sure discussion is relevant to this manuscript or elaborate this.
2. Figure 5 The homogeneity, completeness, AMI, and ARI; author must think about to use contrast colors and line graph with dots can help audience not be lost.

Experimental design

no comment

Validity of the findings

no comment

Additional comments

no comment

·

Basic reporting

The manuscript "Complet+: a computationally scalable method to improve completeness of large-scale protein sequence clustering" proposed a computationally scalable post-processing method, Complet+, for merging clusters and reducing redundancy in formed clusters.
Overall, the manuscript is well written with well-referenced sentences with citations to substantiate them.
However, the manuscript still needs a few minor revisions to be recommended for publication. Some comments/ questions/ suggestions towards that are listed below:
1. For the easy understanding of the target readers, the paper needs a detailed description of runtime and sensitivity analysis for the complete+ algorithm. If possible, the authors are requested to add the graphs for different levels of sensitivity.
2. In the "Methods" section, the authors need to discuss the comparison of estimated unsupervised models in more detail with respect to Complete+. The authors are also advised to add the related graphs in the "Result and Discussion" sections.

Experimental design

No Comment

Validity of the findings

No Comment

Additional comments

No Comment

Reviewer 3 ·

Basic reporting

The manuscript presents a method Complet+ to post-process sequence clusterings in order to improve the completeness (i.e. sensitivity) and homogeneity (i.e. precision) of clusters. The proposed method is tested on clustering of sequences from the SCOP database of protein domains, produced by the clustering tool CD-HIT and various settings of the mmseqs cluster tool. The authors plot the homogeneity, completeness (at family and superfamily level) and combined measures ARI and AMI, for the original clusterings and the ones obtained by post-processing with the proposed method.

Experimental design

no comment

Validity of the findings

Major points:

1. The proposed method, described on lines 235 - 248, is nothing else but connected component clustering, which is implemented in mmseqs as --cluster-mode 1. The reciprocal hit criterion of Complet+ should not have any effect. If sequence X found sequence Y with mmseqs, the reverse should always be true because the raw score of the alignments X-Y and Y-X are the same, and the prefilter of mmseqs should find Y with X whenever X found Y (when X has two similar k-mers in common with Y then Y has two similar k-mers in common with X). Therefore, it is unclear what the contribution of this study is.

2. Complet+ should be compared to Complet+ without the reciprocal hit criterion and to post-processing by "mmseqs clust ... --cluster-mode 1 -s 7.5". All three should do exactly the same and perform the same.

3. mmseqs allows users to compute an MSA and then a sequence profile for each cluster. The all-vs-all search in the final clustering step can then be done using profile-to-sequence search, which is quite a bit more sensitive than sequence-to-sequence search. A work-flow that is therefore very likely to perform better on the SCOP benchmark than Complet+ would use the profile-to-sequence search to create the graph on which compute the final clustering.

4. There are many tools advertised for sequence clustering, for instance Markov clustering. The study should include a comparison a representative selection of the most popular of these tools.

5. SCOP only contains single-domain sequences. When clustering normal, full-length sequences, transitive connections can link sequences that do not share any homologous regions, for instance A is similar to AB (two domains) is similar to B, but A should *not* be clustered with B. The real challenge for sequence clustering tools to deal well with such real-world sequence sets. It requires thinking about minimum sequence coverage criteria and the like. A second, more realistic benchmark with full-length sequences would therefore be needed. Please consult the literature on ideas how to do this.


Minor points:

6. The visualizations of the cluster qualities are hard to read and interpret. Please consult the literatures for ideas to improve this and revise.

7. Why is Default completeness (light blue) missing in columns 1 and 3 of Fig. 5?

---

## Round 0.2 · accepted · Accept

I am pleased to inform you that your manuscript titled "Complet+: a computationally scalable method to improve the completeness of large-scale protein sequence clustering" has been accepted for publication in our journal.

We have carefully reviewed your manuscript and the responses to the comments and queries raised by the reviewers. We are satisfied that the revisions made to the manuscript address all of the points raised and believe that the manuscript is now ready for publication.

We would like to congratulate you on the acceptance of your manuscript and look forward to working with you to bring it to publication.

Thank you for choosing to submit your work to our journal.
Yours Sincerely,
Mahesh